# Successful Clearance of 300 Day SARS-CoV-2 Infection in a Subject with B-Cell Depletion Associated Prolonged (B-DEAP) COVID by REGEN-COV Anti-Spike Monoclonal Antibody Cocktail

**DOI:** 10.3390/v13071202

**Published:** 2021-06-23

**Authors:** Arnaud C. Drouin, Marc W. Theberge, Sharon Y. Liu, Allison R. Smither, Shelby M. Flaherty, Mark Zeller, Gregory P. Geba, Peter Reynaud, W. Benjamin Rothwell, Alfred P. Luk, Di Tian, Matthew L. Boisen, Luis M. Branco, Kristian G. Andersen, James E. Robinson, Robert F. Garry, Dahlene N. Fusco

**Affiliations:** 1Department of Medicine, School of Medicine, Tulane University, New Orleans, LA 70112, USA; sliu26@tulane.edu (S.Y.L.); sflaherty@tulane.edu (S.M.F.); preynau@tulane.edu (P.R.); wrothwel@tulane.edu (W.B.R.); aluk@tulane.edu (A.P.L.); dfusco@tulane.edu (D.N.F.); 2Department of Immunology and Microbiology, Tulane University, New Orleans, LA 70118, USA; mtheberge@tulane.edu; 3Department of Microbiology and Immunology, Tulane University School of Medicine, New Orleans, LA 70112, USA; asmither@tulane.edu (A.R.S.); rfgarry@tulane.edu (R.F.G.); 4The Scripps Research Institute, San Diego, CA 92037, USA; zellerm@scripps.edu (M.Z.); andersen@scripps.edu (K.G.A.); 5Regeneron Pharmaceuticals, Inc., Tarrytown, NY 10591, USA; gregory.geba@regeneron.com; 6Department of Pathology, Tulane University School of Medicine, New Orleans, LA 70112, USA; dtian2@tulane.edu; 7Zalgen Labs, Germantown, MD 20876, USA; mboisen@zalgenlabs.com (M.L.B.); lbranco@zalgenlabs.com (L.M.B.); 8Department of Pediatrics, Tulane University School of Medicine, New Orleans, LA 70112, USA; jrobinso@tulane.edu

**Keywords:** B-DEAP COVID-19, B-cell depletion associated prolonged COVID-19, COVID-19, SARS-CoV-2 persistence, virus mutations, anti-CD20-mediated B-cell depletion, obinutuzumab, REGEN-COV, REGN10933 and REGN10987, spike mutation, anti-COVID-19 vaccine

## Abstract

A 59-year-old male with follicular lymphoma treated by anti-CD20-mediated B-cell depletion and ablative chemotherapy was hospitalized with a COVID-19 infection. Although the patient did not develop specific humoral immunity, he had a mild clinical course overall. The failure of all therapeutic options allowed infection to persist nearly 300 days with active accumulation of SARS-CoV-2 virus mutations. As a rescue therapy, an infusion of REGEN-COV (10933 and 10987) anti-spike monoclonal antibodies was performed 270 days from initial diagnosis. Due to partial clearance after the first dose (2.4 g), a consolidation dose (8 g) was infused six weeks later. Complete virus clearance could then be observed over the following month, after he was vaccinated with the Pfizer-BioNTech anti-COVID-19 vaccination. The successful management of this patient required prolonged enhanced quarantine, monitoring of virus mutations, pioneering clinical decisions based upon close consultation, and the coordination of multidisciplinary experts in virology, immunology, pharmacology, input from REGN, the FDA, the IRB, the health care team, the patient, and the patient’s family. Current decisions to take revolve around patient’s follicular lymphoma management, and monitoring for virus clearance persistence beyond disappearance of REGEN-COV monoclonal antibodies after anti-SARS-CoV-2 vaccination. Overall, specific guidelines for similar cases should be established.

## 1. Introduction

COVID-19 clinical presentation can vary in severity and duration of infection. Most therapeutic options and clinical trials are focused on subjects early in the course of infection. Patients with prolonged acute infection, often associated with immune depletion, have limited therapeutic options. Here, we present a case of depletion associated prolonged (DEAP) COVID-19 treated with the off-label use of synthetic monoclonal antibody.

### Case Report Presentation

A 59-year-old male with obesity (BMI 28.6 kg/m^2^), hypertension, and hypothyroidism was admitted with COVID-19. The hypertension was mild, and the patient was not being treated with any medications (i.e., no angiotensin-converting enzyme inhibitors, nor angiotensin-receptor blockers). Three years prior to admission, the patient had a classical Hodgkin’s lymphoma, grade IV, followed 18 months later by a follicular lymphoma (FL), grade III. Thirteen months prior to writing he received two of five cycles of G-benda, a combination of obinutuzumab, an anti-CD20 B-cell-depleting monoclonal antibody, and bendamustine, an ablative chemotherapeutic agent. Within weeks of the completion of the second G-benda cycle, the patient was admitted to an outside institution for roughly one week of fever, cough, and shortness of breath. Nasal swab qRT-PCR, abnormal imaging, and oxygen desaturation confirmed symptomatic COVID-19 infection. Throughout the month following admission, the patient required high-flow oxygen, however, the patient did not develop acute respiratory distress syndrome (ARDS) or require intubation. Anti-COVID-19 therapy consisted of hydroxychloroquine (400 mg once day one of admission, then 200 mg twice daily on days 2, 3, 7, and 8 of admission), azithromycin (500 mg once daily for the first five days of admission), lopinavir/ritonavir (200/50 mg tablets, two tablets bid days six through 15 of admission), steroids (solumedrol 80 mg iv bid, then 60 mg iv bid days 11 and 12 of admission, respectively), and two infusions of COVID-19 convalescent plasma, 30 and 50 days after diagnosis. All treatments were unsuccessful in clearing virus, as measured by unchanged nasal SARS-CoV-2 qRT-PCR and persistent symptoms. After 6 weeks, the patient was transferred to our center to receive remdesivir. Remdesivir was administered as 200 mg iv once daily for one dose, then 100 mg iv once daily for nine days. Following remdesivir, the patient exhibited slow clinical improvement. Three and a half months following diagnosis, after an overall mild clinical course and decreased oxygen dependence, patient was discharged home to quarantine for an indefinite period, given persistent detectable SARS-CoV-2 by nasal swab qRT-PCR (Appendix A). After 10 weeks at home, he was readmitted for severe consolidated pneumonia, in the context of lympho-neutropenia and persistently low serum IgG (Appendix A), the latter prompting treatment with IVIG. During the week following readmission, he tested negative for SARS-CoV-2 on nasal swabs taken over two consecutive days (COVID-19 IDnow (Abbott)). The pneumonia was treated with broad-spectrum antibiotic/antifungal coverage including piperacillin-tazobactam, vancomycin, voriconazole, and steroids (prednisone 60 mg po once daily). However, three weeks from readmission, since no organism could be isolated and both cough and dyspnea persisted, COVID-19-related pneumonia was suspected; a nasal swab was administered (IDnow) and tested positive. COVID-19 was reconfirmed on subsequent nasal swabs, all with a detectable genome of SARS-CoV-2 by both tests (IDnow, cobas (Roche)). In retrospect, all samples collected within a couple of days of enrollment for the research study (nasal swab, saliva, and residual broncho-alveolar lavage (BAL) fluid) had detectable SARS-CoV-2 by TaqPath/CDC qRT-PCR, ruling out a SARS-CoV-2 re-infection, and presenting more consistently with a single prolonged infection with two false-negative nasal swabs upon readmission. During his subsequent prolonged (6 month) hospitalization, the patient experienced an unchanged clinical status with no improvement but no severe worsening, and mild oxygen dependence requiring on average 4 L/min. Weekly SARS-CoV-2 qRT-PCRs either by IDnow or cobas were all positive (Appendix A). The patient did not develop anti-SARS-CoV-2 humoral response on serial plasma collected from days 80, 210, and 270 following initial diagnosis, using two independent ELISA platforms, both of which detect antibodies against SARS-CoV-2 nucleocapsid protein (NP) and either trimeric spike or spike receptor-binding domain (RBD) (Zalgen and in-house ELISA test) (Appendix A). Sequencing of the virus was possible on nasal swab, saliva, and BAL collected 6 days after re-admission, with sample SARS-CoV-2 Ct values < 14. Sequence analysis of samples collected days 74 to 296 from initial diagnosis showed the persistent accumulation of new mutations over time, resulting in six amino acid changes, some not previously described (Figure 1, GISAID accession numbers: Virus name Accession ID: hCoV-19/USA/LA-ALSR-5820/2020 EPI_ISL_878549, hCoV-19/USA/LA-ALSR-5826/2020 EPI_ISL_878563, hCoV-19/USA/LA-ALSR-6526/2021 EPI_ISL_962812, hCoV-19/USA/LA-ALSR-6528/2021 EPI_ISL_962813, hCoV-19/USA/LA-ALSR-6529/2021 EPI_ISL_962814, hCoV-19/USA/LA-ALSR-6531/2020 EPI_ISL_962815). Eight months following initial symptom onset, alerted by report of variants with increased transmissibility/severity [1], and having concerns regarding the possible accumulation of new mutations, we proposed to attempt to clear virus carriage using an anti-SARS-CoV-2 spike protein monoclonal antibody cocktail, casirivimab/imdevimab (REGEN-COV).

## 2. Materials and Methods

### 2.1. Sample Processing Protocols

Plasma prepared from anticoagulated fresh blood draw and residual samples are frozen and stored at −80 °C in aliquots of 200 µL.

### 2.2. Serology Tests

#### 2.2.1. ReSARS^®^ CoV-2 N-Protein and S-RBD IgG ELISA Kits for RUO, Zalgen Labs, LLC

ReSARS^®^ CoV-2 N-protein and S-RBD IgG ELISA kits for RUO, Zalgen Labs, LLC (Germantown, MD, USA) are formatted as direct ELISAs with recombinant antigens coated directly on to 96-well microtiter plates (Nunc) and stabilized with proprietary block solution. The assays are semi-quantitative with convalescent COVID-19 donor plasma (lyophilized) provided to prepare a reference curve. Additional COVID-19 donor plasma (lyophilized) is provided as a positive control and bulk normal human plasma (lyophilized) is provided as a negative control. After reconstituting the reference and controls with purified water, the reference, controls, and human plasma samples (K3EDTA) are diluted 1:101 in sample diluent (10:1000 µL). The reference is further diluted 4-fold to prepare a five-point reference curve. The reference curve, controls, and samples are added to duplicate, designated microwells at 100 µL/well and incubated at ambient temperature (18–30 °C) for 30 min. Two wells of just sample diluent are included as a reagent blank. Following incubation, the plates are washed 4 × 300 µL/well with PBS-Tween wash buffer. For detection of N-protein or S-RBD-specific IgG, affinity purified goat anti-human IgG HRP conjugate reagent is added to the microwells at 100 µL/well and incubated at ambient temperature for 30 min. PBS-Tween wash is repeated to remove conjugate reagent. TMB substrate (Moss) is added to microwells (100 µL/well) for 10 min at ambient temperature. Substrate color development is stopped with the addition of 2% methane sulfonic acid Stopping Solution. Microwell plates are read at 450 nm with 650 nm reference. Gen5 program is used to subtract mean 450–650nm followed by reagent blank subtraction. Specific IgG (units/mL) is calculated from a 4-parameter logistic fit of the reference curve. The valuation of the reference plasma is based on a 0–100 scale with 100 units/mL equivalent to maximum clinical titer. Negative cut-off is equivalent to 2 units/mL based on mean + 2.5× Standard Deviation of +100 pre-COVID US normal plasmas.

#### 2.2.2. SARS-CoV-2 Anti-S (HEXAPRO) ELISA for RUO (Dr Robinson’s Lab)

A plasmid for the expression of stable pre-fusion trimeric SARS-CoV-2 spike protein, similar to that described by Wrapp et al [2], was kindly provided by Kate Hastie (La Jolla Institute for Immunology). Recombinant stabilized spike protein is produced and purified as described by [3] Dan et al. Wells of 96-well ELISA plates (Costar, Easy Wash) are coated for 1 h at room temperature with this SARS-CoV-2 spike protein (500 ng/well in 100 mM sodium bicarbonate buffer). Wells are washed 5× and blocked for 1 h with 0.5%Tween, 5% dry milk, 4% whey, 10%FBS in 1× PBS at 37 °C. Sera or plasma samples diluted to 1:100 in the same buffer are incubated in antigen-coated and uncoated wells for 1 h at room temperature. Bound IgG is detected with peroxidase-conjugated goat anti-IgG (Jackson Immunoresearch, West Grove, PA, USA) and color is developed with TMB-peroxidase as described [4]. Net OD values at 450 nm were calculated by subtracting background OD readings from OD readings with spike protein. A cut off net OD value of 0.451 is calculated based on testing of >100 pre-COVID-19 serum samples.

#### 2.2.3. COVID-19 ACE2 Competition Assay^®^ (V-PLEX SARS-CoV-2 Panel 2), Meso Scale Discovery, for RUO

This multiplexed solid-phase chemi-luminescence assay allows the simultaneous detection of IgG binding to four SARS-CoV-2 antigens and the quantification of antibody-induced ACE-2 binding inhibition. The latter is an ELISA-based pseudo-neutralization assay, detecting antibodies able to block the binding of angiotensin-converting enzyme 2 (ACE2) to the SARS-CoV-2 spike and S1 RBD antigens. Plates are blocked and washed, serial dilution of assay calibrator (COVID-19-neutralizing antibody; monoclonal antibody against S protein; 200 unit/mL), control and plasma samples (10 μL) diluted 1:100 in assay diluent are added to the plates. Following sample incubation an 0.25 μg/mL solution of MSD SULFO-TAG™ conjugated ACE-2 is added, after which plates are read. The assay is performed with the Meso QuickPlex SQ120 on a 10-spot 96-well microplate. Data is extrapolated from a calibration curve and calculated as percent inhibition normalized by a blank sample or as concentration of neutralizing SARS-CoV/SARS-CoV-2 spike monoclonal antibody. The value of 1 unit/mL of concentration of calibrator corresponds to neutralizing activity of 1 μg/mL monoclonal antibody to SARS-CoV/SARS-Cov-2 spike protein.
% inhibition = [(1 − average sample single)/average single of Calibrator 8] × 100

### 2.3. Molecular Tests

#### 2.3.1. CDC Emergency Use Authorization (EUA) 2019-nCoV Real-Time qRT-PCR Panel Protocol, A. Smither and Dr. Garry’s Lab. for RUO

Samples: Research swabs, saliva samples, and BAL collected prior to REGN infusion are processed with this qRT-PCR protocol. Sample RNA extraction is performed with QIAmp Viral RNA Mini Kit (Qiagen) according to the manufacturer’s instructions. Primers: The presence of SARS-CoV-2 nucleocapsid (NP) is assessed using 2 sets of primers designed to target two portions of the NP gene, and a human gene RNAseP (RP) (Table 1).

Reaction mix: 8.5 μL nuclease-free water, 1.5 μL combined primer/probe mix (Integrated DNA Technologies), 5 μL 4× TaqPath 1-Step RT-qPCR Master Mix, CG (Applied Biosystems), and 5 μL of RNA (extract or standard) in a final volume of 20 µL. Each primer/probe set is run as a singlet for a total of three reactions/specimen. Amplification: Cycling parameters programed as 25 °C for 2 min, 50 °C for 15 min, 95 °C for 2 min, and 40 cycles of 95 °C for 3 s and 55 °C for 30 s, in a QuantStudio 3 Real-Time PCR System (Applied Biosystems). Calibrator: Viral load is assessed using an N1-specific ssRNA standard dilution series ranging from 10^8^–10^1^ copies (kind gift of Chris. Monjure). Analysis: A specimen is considered positive if all targets, viral (N1; N2) and human (RP), have a Ct < 35.

#### 2.3.2. Applied Biosystems TaqPath COVID-19 Combo Kit (Dr. Tian’s Lab) for IVD Use

Research swabs collected immediately prior to and after REGN MoAb infusions were processed in the Molecular Laboratory of the Tulane Department of Pathology (Dr. Tian, MD., PhD. Director). The laboratory is certified under CLIA and accredited by CAP to perform high-complexity testing. The test is approved under FDA Emergency Use Authorization (EUA). RNA extraction is performed using the KingFisher Flex automated extractor. Extraction requires a starting sample volume of 200 µL of the swab initial material and 10 µL of the final 50 µL elution volume is used per RT-PCR reaction. The SARS-CoV-2 RT-PCR is performed with the Thermo-Fisher TaqPath COVID-19 Combo Kit following the manufacturer’s recommendations. For amplification, we use the QS5 QuantStudio Real-time PCR system, with a total of 40 cycles. A cut-off Ct value set at 37 cycles is used to set the detection of the viral genes N, ORF1ab and S.

#### 2.3.3. SARS-CoV-2 Genome Sequencing (Andersen Lab. Scripps Research Institute), for RUO

Samples with an N1 Ct < 30 (correlating to ~500 copies of virus/μL) were selected for amplicon sequencing. Samples’ viral RNA was shipped to Scripps Research Institute. All samples passed the required QC, consisting of RNA concentration measurement (Qubit) and fragment size evaluation (TapeStation) before sequencing. Virus sequencing is performed using NextSeq. For analysis, sequence reads are aligned to a hCoV-19 reference sequence and consensus sequences generated using iVar.

## 3. Results

The compassionate use of emergency IND was approved by the FDA and the local IRB. Virus mutation analysis was consistent with persistent infection, and the lack of expected casirivimab/imdevimab escape: 1—one strain of virus, inferring that the virus was of identical strain before and after discharge home/readmission; 2—the constant accumulation of mutations within S and N sequences; 3—the absence of mutations that would decrease the efficiency of anti-S monoclonal antibody (Figure 1).

On day 270 the patient received the first infusion of REGEN-COV anti-spike monoclonal antibody cocktail at a dose of 2.4 g. Over subsequent weeks, 10 successive nasal swabs and 13 nasopharyngeal swabs were performed to monitor effects on viral persistence (Figure 2 and Figure 3). Tested by IDnow, after two detectable swabs, the virus became undetectable four times. However, all samples tested by cobas remained detectable. Samples tested by TaqPath immediately following infusion had Ct~13, followed by increase of all Ct values to a range of 28 and up to 30 with increased time post-infusion. We concluded that viral clearance was incomplete. Since the mean estimated half-life for both antibodies in casirivimab/imdevimab ranges from 25 to 37 days [5], a consolidation infusion was administered 35 days (~6 weeks) after first infusion, at a higher dose of 8 g (4 g for each of casirivimab and of imdevimab, respectively). Serological testing after infusion confirmed the absence of anti-N antibodies and the presence of anti-S only with neutralizing effect confirmed by Mesoscale discovery COVID-19 ACE2 competition assay, with results ranging from absent at baseline to >100% after first and second infusions (Appendix A). After a second monoclonal antibody infusion, the results of all swabs, performed bi-weekly with all three techniques (IDnow, cobas, TaqPath), were non-detectable. Over the period of infusions, days 250–330, the patient underwent a long, well-tolerated, steroid taper (from 60 mg once daily to no steroids, tapered down by 5 mg roughly every week over twelve weeks). After 360 days, the patient was noted to have persistent pancytopenia, despite what appeared to be SARS-CoV-2 virus clearance; therefore, a bone marrow biopsy was performed, revealing no involvement by the patient’s FL. Peripheral blood immunophenotyping (Appendix A) showed virtual absence of CD20 B-cells, low CD4 and CD8 T-cells, and persistent low serum IgG levels (Appendix A), all suggesting continued effects of G-benda. Since the expected recovery of some B-cell functionality was reached, allowing more reasonable chances for vaccine success, and in absence of clear contraindication, the patient received a first dose of Pfizer-BioNTech COVID-19 vaccine, 390 days following initial diagnosis. He was then discharged to home, where he would live with one family member (who had received vaccine dose one), with plans for close clinical follow up and regular check-ins from the state department of health. 

## 4. Discussion

Concerns have been raised that patients with cancer, treated with ablative chemotherapy and B-cell depletion mediated by anti-CD20 monoclonal antibody, may experience more severe COVID-19 infection and a decreased efficiency of the anti-SARS-CoV-2 vaccine [6,7]. Since anti-CD20-MoAb-mediated B-cell depletion is prolonged (6–9 months), and normal B-cell levels are only reached after ~12 months [8], concerns related to inadequate humoral immune response are legitimate. In this patient, anti-SARS-CoV-2 antibodies were tested 80, 95, 210, and 220 days after diagnosis, using two independent ELISAs for the detection of the main virus antigens, nucleocapsid, spike protein in its trimeric form, and spike receptor-binding domain. All pre-REGEN-COV plasma showed no reactivity to virus antigens, which is consistent with the absence of neutralizing activity tested at 210 and 220 days (Appendix A). More than one year after the last dose of obinutuzumab/bendamustine, our patient had persistent and pronounced lymphopenia, hypogammaglobulinemia (IgG~200 mg/dL), and no CD20/low CD4 T-cells in peripheral blood by flow cytometry. Of note, bendamustine can induce a very delayed T-lymphocyte reconstitution, particularly of CD4+ T cells [9], raising the possibility that benda alone or benda enhanced by COVID-19 was suppressing this patient’s CD4+ T cells. This case report is a striking illustration that anti-CD20-mediated B-cell depletion is not always associated with severe or life-threatening COVID-19 [10], but also with a mild yet prolonged course, as reported [6,11]. In this regard, non-specific immunity and the specific non-B-cell-dependent host defenses adequately contained the infection, or at least prevented the destructive effects of massive virus replication. Consistently, NK-cell function has even been reported as increased upon the anti-CD20 depletion of B-cells [12]. The absence of humoral response may also have prevented its deleterious side effects, such as antibody-dependent cell cytotoxicity or excessive inflammatory response [6,13]. In the absence of specific natural humoral response and after the ineffective immune pressure of multiple convalescent plasma infusions, the acquisition/shedding of variants of increased virulence or with the potential for immune escape was eventually our main concern [1]. Serial sequencing showed that the spike region identified no definitive casirivimab/imdevimab escape mutants, so the specific neutralizing effect of casirivimab/imdevimab toward the patient’s variants was predicted [14]. To optimize patient care, we firstly implemented drastic confinement measures with concern for virus mutation in a patient with active shedding, and secondly, sequenced isolates from respiratory samples over time to monitor strain mutations and predict the efficacy of the anti-spike REGEN-COV antibody cocktail.

Other cases of prolonged COVID following the use of anti-CD20-depleting agents have been reported in the literature, including one case where repeated administration of convalescent plasma was associated with large and dynamic virus population shifts, without virus clearance [15]. Similarly, the current case reports B-DEAP COVID-19 with lack of virus clearance upon treatment with convalescent plasma. In contrast however, this case describes subsequent successful virus clearance using a synthetic anti-SARS-CoV-2 cocktail. It is hoped that the case described here might provide the glimmer of a potential treatment avenue to be pursued more systematically for future B-DEAP-COVID-19 cohorts.

Related to the performance of diagnostics in this case, in two instances the analytical sensitivity of IDnow appeared to be lower than that of cobas or TaqPath. First, at the time of re-admission for pneumonia, the question of persistent SARS-CoV-2 infection was challenged, as two immediate consecutive swabs were negative by IDnow. In contrast, qRT-PCR in saliva, nasal swab, and BAL had detectable amounts of virus, arguing that the patient had an ongoing, continuous infection. Second, after the first REGEN-COV infusion, testing for SARS-CoV-2 was repeated in nasal swabs by IDnow and cobas (Figure 2). SARS-CoV-2 was reproducibly not detected by IDnow but detected by cobas. The discrepancy has led us to suggest the partial success of the first REGEN-COV infusion, also supported by repeatedly higher Ct values of TaqPath assay performed after the first infusion. Notably, we could not perform a strict correlation of Ct values and viral load in the absence of an internal genomic calibrator in the samples. Altogether, it is important to recall that test sensitivity/specificity and test clinical value are two separate notions, depending primarily upon the clinical question that needs to be answered. In clinical cases such as the one described here, pending clearer guidelines, the authors would suggest that clinicians employ multiple distinct SARS-CoV-2 diagnostic tests when clinical suspicion is high, and sequence the virus/publicly share the diagnostic test and sequence results whenever possible.

Anti-SARS-CoV-2 vaccine was administered in the window where REGEN-COV antibody cocktail levels were expected to decline, and when we could expect an effective vaccine response regarding B-cell-depletion recovery. This usually occurs 11 months after the last anti-CD20 dose, even in the absence of B cells detected in peripheral blood. We are still unable to predict whether our patient will efficiently develop a vaccine response, and we will follow up with serial visits, a complete blood count, serum IgG levels, anti-S Ab titers, neutralization assay, and nasal swabs. It is our hope that future studies can be performed to evaluate for the presence of activated T-cells or SARS-CoV-2-specific T-cells. Persistent pancytopenia and hypo-gamma-globulinemia still observed one year after two cycles of G-benda have not yet been resolved. First, they are likely attributable to G-benda since complete blood count parameters and IgG levels were normal prior to these treatments. Second, they are not associated with relapsed/persistent FL involvement by recent bone marrow biopsy study. Additional marrow-suppressive effects of SARS-CoV-2 cannot be excluded. 

Obinutuzumab is a second-generation humanized anti-CD20 monoclonal antibody that has been developed to increase complement-dependent cytotoxicity and/or antibody-dependent cellular cytotoxicity by enhancing binding affinity for the Fc-γ receptor III expressed on immune effector cells. Early and late-onset neutropenia have been reported with greater incidence in obinutuzumab compared to rituximab (GAZYVA full Prescribing Information. Genentech, Inc.; 2020). Hypo-gammaglobulinemia was reported as long as 6 years after rituximab anti-CD20 B-cell depleting therapy, a rare phenomenon possibly associated with a polymorphism of the gene coding for FcγR3 [16]. SNPs *FCGR2A*[1 31H/R] and *FCGR3A*[158F/V] [17] have both been associated with increased affinity of Fcγ receptors on lymphocytes and neutrophils, and, if present, could explain prolonged cytopenia and hypogammaglobulinemia. Evaluation for these FC-gamma receptor (FCGR) variants has not been performed but should be considered, as should additional immune-function assays. Of note, since obinutuzumab recently received FDA approval, in contrast to rituximab, the drug’s long term side effects on white blood cells and IgG levels remain anecdotal in the literature [18,19]. The patient had a hypothyroidism of autoimmune origin and an exceedingly rare sequence of classical Hodgkin’s lymphoma followed by a high-grade follicular lymphoma [20] within a couple of years, raising concerns for congenital immunodeficiency and autoimmune lymphoproliferative disorder [21]. In this regard, we could rule out common variable immunodeficiency with normal IgG and IgA levels prior to G-benda, and normal complement CH50, C3, and C5.

## 5. Conclusions

This is the first case of anti-CD20-mediated B-cell depletion associated prolonged COVID-19 (B-DEAP COVID-19) treated successfully with a clearance of ~300 days of SARS-CoV-2 infection by REGEN-COV monoclonal antibody cocktail infusion. Specific guidelines should be developed on how to manage infection control, monitor virus genetic mutations, and best provide passive (SARS-CoV-2 specific antibody infusions) and active immunizations in the case of B-DEAP COVID-19.

## Figures and Tables

**Figure 1 viruses-13-01202-f001:**
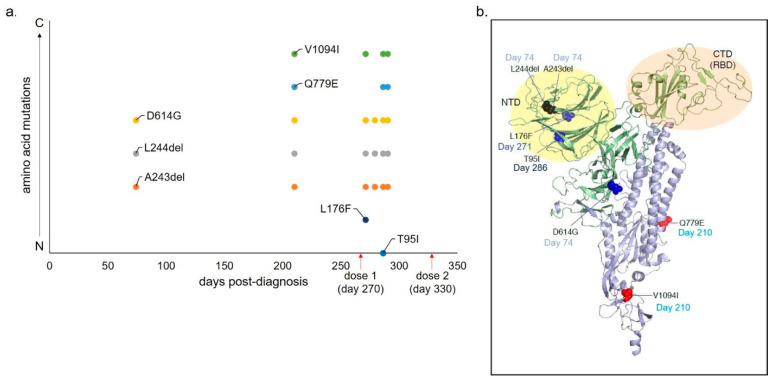
Amino acid mutations detected over time through serial SARS-CoV-2 sequencing during 300 days of virus shedding in patient with B-DEAP (B-cell depletion associated prolonged) COVID-19. (**a**) X-axis indicates days post-diagnosis. Y-axis indicates amino acid mutations detected through the sequencing of nasal swabs positive for SARS-CoV-2 RNA with Ct value < 30, over time. (**b**) Structural representation of mutations over time.

**Figure 2 viruses-13-01202-f002:**
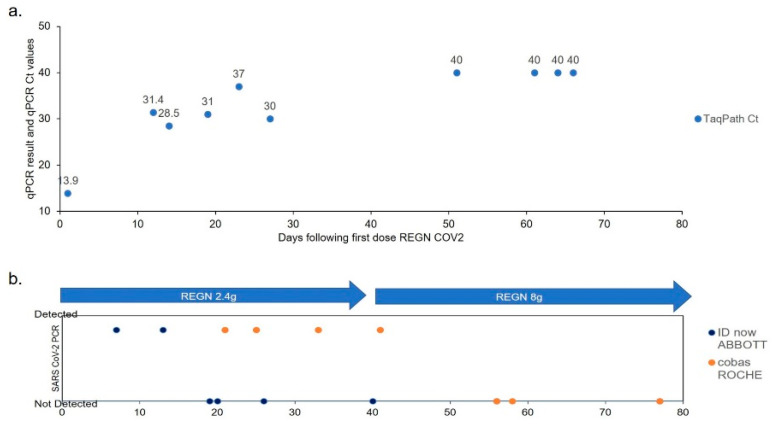
SARS-CoV-2 qRT-PCR results following REGEN-COV anti-spike MoAb infusions. (**a**) TaqPath Ct values. (**b**) IDnow and cobas Roche results. First dose infusion day 0. TaqPath Ct < 37 detected; Ct > 37 not detected. IDnow and cobas not detected vs. detected.

**Figure 3 viruses-13-01202-f003:**
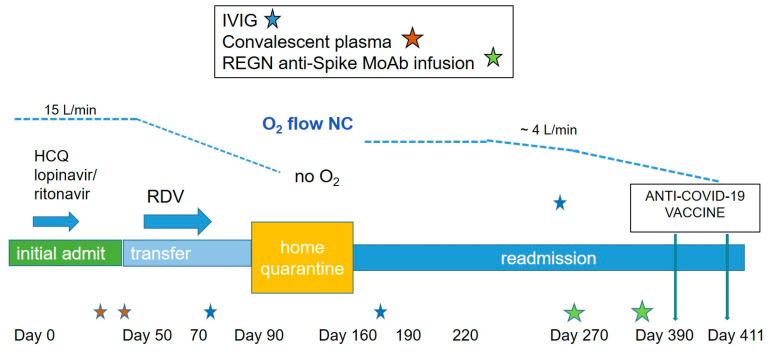
Clinical course, therapies, virus detection. Chronological presentation of patient’s course, including supplemental oxygen required, therapeutic interventions, anti-SARS-CoV-2 antibody results, and SARS-CoV-2 PCR results. NC = nasal cannula. HCQ = hydroxychloroquine. RDV = remdesivir. L/min = liters per minute. IVIG = intravenous immune globulin.

**Table 1 viruses-13-01202-t001:** Primers.

Primer/Probe Mix	Sequence	
2019-nCoV_N1	Forward	5′-GACCCCAAAATCAGCGAAAT-3′
	Reverse	5′-TCTGGTTACTGCCAGTTGAATCTG-3′
	Probe	5′-FAM-ACCCCGCAT/ZEN/TACGTTTGGTGGACC-3IABkFQ-3′
2019-nCoV_N2	Forward	5′-TTACAAACATTGGCCGCAAA-3′
	Reverse	5′-GCGCGACATTCCGAAGAA-3′
	Probe	5′-FAM-ACAATTTGC/ZEN/CCCCAGCGCTTCAG-3IABkFQ-3′
Hu RNAseP (RP)	Forward	5′-AGATTTGGACCTGCGAGCG-3′
	Reverse	5′-GAGCGGCTGTCTCCACAAGT-3′
	Probe	5′-FAM-TTC TGA CCT/ZEN/GAA GGC TCT GCG CG-3IABkFQ-3′

## Data Availability

Data supporting the reported results can be obtained by emailing the corresponding author.

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
