# Peer review of "Successful Clearance of 300 Day SARS-CoV-2 Infection in a Subject with B-Cell Depletion Associated Prolonged (B-DEAP) COVID by REGEN-COV Anti-Spike Monoclonal Antibody Cocktail"

_viruses, 2021, doi:10.3390/v13071202_

Round 1
Reviewer 1 Report
This case report has valuable information about viral characteristics and immunity. Even though the administration of REGEN-COV was effective, a few more details need to be elaborated.
Major revision
1. SARS-CoV2 specific CD4/CD8 T cell analysis and the detailed description of the steroid dosage/tapering
Considering the patient's long clinical course and the emergence of variants after the 1st dose of REGEN-CoV, I think there was a possibility of the clearance of viruses by SARS-CoV2 specific ADCC/ or CTL. Therefore, an analysis for COV2-specific CD4/CD8 analysis will give more precise insight into REGEN-COV therapy.
Also, it is needed to describe the information about steroid treatment (esp., during steroid tapering)
Author Response
Reviewer 1
Top of Form
Comments and Suggestions for Authors
This case report has valuable information about viral characteristics and immunity. Even though the administration of REGEN-COV was effective, a few more details need to be elaborated.
Major revision
- SARS-CoV2 specific CD4/CD8 T cell analysis and the detailed description of the steroid dosage/tapering
Considering the patient's long clinical course and the emergence of variants after the 1st dose of REGEN-CoV, I think there was a possibility of the clearance of viruses by SARS-CoV2 specific ADCC/ or CTL. Therefore, an analysis for COV2-specific CD4/CD8 analysis will give more precise insight into REGEN-COV therapy.
** We greatly appreciate the Reviewer’s suggestion. We unfortunately have not performed CD4/8 T cell analyses in this subject, due to limited resources. We have, however, applied for funding to more extensively study such cases, with anti-SARS-CoV-2 CD4/8 analyses in the future and hope that this will become possible, as we are being contacted on a regular basis regarding potential treatment options for other patients with similar situation. We do hope that this manuscript will prompt future creation of guidelines for both diagnostic and treatment management of patients with DEAP-COVID and do fully acknowledge the importance of including CD4/8 analyses moving forward. We regret that we cannot provide that data for this particular case, and hope that the Reviewer understands we do hope, with recognition of this phenomenon (DEAP COVID) additional future studies might become possible.
Also, it is needed to describe the information about steroid treatment (esp., during steroid tapering)
** We thank the Reviewer for this comment. The specific description of medication/dose/taper for steroids have been provided in the clinical case description.
Submission Date
24 April 2021
Date of this review
25 May 2021 08:51:54
Reviewer 2 Report
Comments for Author
This is a very interesting case report that does an excellent job of addressing the challenges faced by the expert team when combining pre-existing conditions with COVID and the complexity of the significance of the analysis results.
There are very few questions and remarks:
1) page 2, line 93, a space after his should be taken away
2) page 4, line 144: vaccination 390 days following initial diagnosis – compare to Figure 3 – complete overview, here it is day 330 and 360?
3) page 6, line 179-181: full sentence, statement: “Altogether it is important….upon clinical question that needs to be answered” is fully correct.
But, with respect to the case, what would be the consequence? Would you recommend in such a complex case to always combine several tests, or to take the test with highest specificity? Then the authors should add their proposal.
4) page 6, line 187: CBC needs to be written out
5) page 6, last line: FCGR (FC-gamma receptor, written out)
6) There is one interesting recent article by Kem et al. (Kemp et al., 2021) which could/should be mentioned and discussed, especially against the background that many such reports are to be collected to enable orientation in the future.
Kemp SA, Collier DA, Datir RP, Ferreira IATM, Gayed S, Jahun A, Hosmillo M, Rees-Spear C, Mlcochova P, Lumb IU, Roberts DJ, Chandra A, Temperton N, Sharrocks K, Blane E, Modis Y, Leigh K, Briggs J, Gils M van, Smith KGC, Bradley JR, Smith C, Doffinger R, Ceron-Gutierrez L, Barcenas-Morales G, Pollock DD, Goldstein RA, Smielewska A, Skittrall JP, Gouliouris T, Goodfellow IG, Gkrania-Klotsas E, Illingworth CJR, McCoy LE, and Gupta RK (2021) SARS-CoV-2 evolution during treatment of chronic infection. Nature 1–10, Nature Publishing Group.
Author Response
Reviewer 2
Comments and Suggestions for Authors
Comments for Author
This is a very interesting case report that does an excellent job of addressing the challenges faced by the expert team when combining pre-existing conditions with COVID and the complexity of the significance of the analysis results.
There are very few questions and remarks:
- page 2, line 93, a space after his should be taken away
** Done, Thank you.
- page 4, line 144: vaccination 390 days following initial diagnosis – compare to Figure 3 – complete overview, here it is day 330 and 360?
** We thank the Reviewer for pointing this out- the vaccine was given Day 390 from initial diagnosis, consistent with the text but not the original Figure 3. Figure 3 has now been corrected to indicate that vaccine doses 1 and 2 were administered at 390 and 411 from initial diagnosis. Thank you for noticing this.
3) page 6, line 179-181: full sentence, statement: “Altogether it is important….upon clinical question that needs to be answered” is fully correct.
But, with respect to the case, what would be the consequence? Would you recommend in such a complex case to always combine several tests, or to take the test with highest specificity? Then the authors should add their proposal.
** We thank the Reviewer for this helpful comment. To address this suggestion, we have added the comment below to the third paragraph of the Discussion:
“In clinical cases such as the one described here, pending clearer guidelines, the authors would suggest that clinicians employ multiple distinct SARS-CoV-2 diagnostic tests when clinical suspicion is high, and sequence virus / publicly share diagnostic test and sequence results whenever possible. “
Also, because our team is now receiving multiple calls on additional DEAP-COVID cases, we will also work to develop some form of data table or publicly shared repository that can continue to address the question over time: how are our available diagnostic tests performing as the virus continues to mutate?
- page 6, line 187: CBC needs to be written out
** Done, Thank you.
- page 6, last line: FCGR (FC-gamma receptor, written out)
** Done, Thank you.
6) There is one interesting recent article by Kem et al. (Kemp et al., 2021) which could/should be mentioned and discussed, especially against the background that many such reports are to be collected to enable orientation in the future.
Kemp SA, Collier DA, Datir RP, Ferreira IATM, Gayed S, Jahun A, Hosmillo M, Rees-Spear C, Mlcochova P, Lumb IU, Roberts DJ, Chandra A, Temperton N, Sharrocks K, Blane E, Modis Y, Leigh K, Briggs J, Gils M van, Smith KGC, Bradley JR, Smith C, Doffinger R, Ceron-Gutierrez L, Barcenas-Morales G, Pollock DD, Goldstein RA, Smielewska A, Skittrall JP, Gouliouris T, Goodfellow IG, Gkrania-Klotsas E, Illingworth CJR, McCoy LE, and Gupta RK (2021) SARS-CoV-2 evolution during treatment of chronic infection. Nature 1–10, Nature Publishing Group.
** Thank you for pointing out this important reference- in response to this comment we have expanded our discussion of COVID and lymphoma / immunodepletion in general, adding to the beginning of our Discussion section two paragraphs, including a discussion of the Kemp 2021 reference.
Reviewer 3 Report
The authors of the manuscript entitled “Successful Clearance of 300 day SARS-CoV-2 Infection in a Subject with B-cell Depletion Associated 4 Prolonged (B-DEAP) COVID by REGN-COV2 Anti-Spike Monoclonal Antibody Cocktail” presented a case report where they succeeded to manage ~300 days of SARS-COV-2 infection in a patient with follicular lymphoma who was treated with G-benda before the infection. By using a combined cocktail of neutralizing anti-spike monoclonal antibodies (REGN-COV2) they show complete viral clearance after the second infusion of 8g Abs. The clinical tests were well designed following the progress of infection, the state of the immune system and virus mutations. Based on this case, they note the need to be developed specific guidelines for treatment of such patients.
Major points:
- The structure of the manuscript should be reorganized: either in a classical way (Introduction, Materials and Methods, Results, Discussion, Conclusion) or as a case study (Introduction, Case report/presentation, Discussion, Conclusion). In the present form there is no Introduction (in this section is given the case presentation) and the methods are given as Supplementary materials. First paragraph of the discussion looks like results.
- In the case presentation (Introduction, line 60) is written that the patient was with hypertension, but there is no information whether any treatment has been given. It is known that taking ACE inhibitors or ARBs could affect the severity of infection. Even there are suggestions that such drugs can help against COVID-19. Also, for the performed anti-COVID-19 therapy (lines 70-73), the used drugs are given, but there is no information for the doses and the treatment period.
- Received sequences have to be deposited in a database. Accession numbers of the sequences are not given in the manuscript.
Minor points:
- In many places in the text, the antibody cocktail is given as REGEN-COV instead REGN-COV2.
- Lines 116, 218 are used terms “phylogenetic analysis” and “evolution”. I do not think that in this case we can talk about phylogeny and evolution. We can talk only about mutations.
- References 3-12 from the list are not cited anywhere.
- Authors detected absence of B-cells and low levels of T-cells suggesting presence of common variable immunodeficiency. It will be interesting to show (if they have performed such FACS analysis) whether there are activated T-cells or SARS-COV-2 specific T-cells. At least to check for such cells after the vaccination (this is not planned – line 187). Otherwise, the vaccination will be pointless.
Author Response
Reviewer 3
Top of Form
Comments and Suggestions for Authors
The authors of the manuscript entitled “Successful Clearance of 300 day SARS-CoV-2 Infection in a Subject with B-cell Depletion Associated 4 Prolonged (B-DEAP) COVID by REGN-COV2 Anti-Spike Monoclonal Antibody Cocktail” presented a case report where they succeeded to manage ~300 days of SARS-COV-2 infection in a patient with follicular lymphoma who was treated with G-benda before the infection. By using a combined cocktail of neutralizing anti-spike monoclonal antibodies (REGN-COV2) they show complete viral clearance after the second infusion of 8g Abs. The clinical tests were well designed following the progress of infection, the state of the immune system and virus mutations. Based on this case, they note the need to be developed specific guidelines for treatment of such patients.
Major points:
- The structure of the manuscript should be reorganized: either in a classical way (Introduction, Materials and Methods, Results, Discussion, Conclusion) or as a case study (Introduction, Case report/presentation, Discussion, Conclusion). In the present form there is no Introduction (in this section is given the case presentation) and the methods are given as Supplementary materials. First paragraph of the discussion looks like results.
** We thank the Reviewer for this comment. We have reformatted according to this suggestion, adding a brief Introduction, moving the case report to a section titled “Case Report Presentation”, eliminating the section “Methods” (which are still available, and listed, as a Supplement), reformatted the “Discussion” section to include clearer discussion, and maintained the “Conclusion” section.
In the case presentation (Introduction, line 60) is written that the patient was with hypertension, but there is no information whether any treatment has been given. It is known that taking ACE inhibitors or ARBs could affect the severity of infection. Even there are suggestions that such drugs can help against COVID-19. Also, for the performed anti-COVID-19 therapy (lines 70-73), the used drugs are given, but there is no information for the doses and the treatment period.
** We thank the Reviewer for these comments. Information about hypertension has been added to the Case Discussion as follows “The hypertension was mild and the patient was not being treated with any medications (ie, no angiotensin converting enzyme inhibitors nor angiotensin receptor blockers).”
Related to treatments given at initial (outside) hospital for COVID-19, we have added the dose and duration information requested to the first paragraph of the Case Report Discussion,
“Anti-COVID-19 therapy consisted of hydroxychloroquine (400 mg once day one of admission, then 200 mg twice daily on days 2, 3, 7 and 8 of admission), azithromycin (500 mg once daily for first five days of admission), lopinavir/ritonavir (200/50 mg tablets, two tablets bid day six through 15 of admission), steroids (solumedrol 80 mg iv bid then 60 mg iv bid days 11 and 12 of admission, respectively) and two infusions of COVID-19 convalescent plasma, 30 and 50 days after diagnosis, all of which were unsuccessful in clearing virus as measured by unchanged nasal SARS-CoV-2 qRT-PCR and persistent symptoms.”
For the steroid taper, we have added clarification:
Over the period of infusions, day 250-330, the patient underwent a long, well tolerated steroid taper (from 60 mg once daily to no steroids, tapered down by 5 mg roughly every week over twelve weeks).
- Received sequences have to be deposited in a database. Accession numbers of the sequences are not given in the manuscript.
We thank the Reviewer for pointing this out- the accession numbers, listed below, have been added to the Manuscript, Case Report Presentation section, :
Virus name Accession ID
hCoV-19/USA/LA-ALSR-5820/2020 EPI_ISL_878549
hCoV-19/USA/LA-ALSR-5826/2020 EPI_ISL_878563
hCoV-19/USA/LA-ALSR-6526/2021 EPI_ISL_962812
hCoV-19/USA/LA-ALSR-6528/2021 EPI_ISL_962813
hCoV-19/USA/LA-ALSR-6529/2021 EPI_ISL_962814
hCoV-19/USA/LA-ALSR-6531/2020 EPI_ISL_962815
Minor points:
- In many places in the text, the antibody cocktail is given as REGEN-COV instead REGN-COV2.
** We thank the Reviewer for this comment. The current name for EUA approved dose of 2.4 grams (1.2 grams each of casirivimab and imdevimab) is REGEN-COV, so the authors have now corrected the manuscript to use this name only, and consistently. The second 8 gram dose we refer to as a higher dose of the antibody cocktail: 4 grams each of casirivimab and of imdevimab, respectively.
Lines 116, 218 are used terms “phylogenetic analysis” and “evolution”. I do not think that in this case we can talk about phylogeny and evolution. We can talk only about mutations.
**The authors greatly appreciate this comment. The terms “phylogenetic analysis” and “evolution” have been removed and replaced with discussion of mutations.
- References 3-12 from the list are not cited anywhere.
** We apologize for this error - we have extended our discussion slightly and these references are now cited.
- Authors detected absence of B-cells and low levels of T-cells suggesting presence of common variable immunodeficiency. It will be interesting to show (if they have performed such FACS analysis) whether there are activated T-cells or SARS-COV-2 specific T-cells. At least to check for such cells after the vaccination (this is not planned – line 187). Otherwise, the vaccination will be pointless.
** The authors very much agree that T cell activation / SARS-CoV-2 specific T cell studies would be helpful. While we have not performed these studies, due to limited resources, we have applied for funding to hopefully permit such studies in the future. We have, since the submission of this case report, been requested to help evaluate several other similar cases and are working to fund / develop a systematic approach to these cases and T cell functional assays, pre- and post-antibody (if we can make antibody available) will be proposed. We do hope that publication of this report will draw further attention to the need for active research in this area of DEAP COVID, and more regular access to antibody infusions for these patients. We very much appreciate the Reviewer’s comment. In direct response to the Reviewer, we have also added to the Discussion the comment below:
“It is the hope of the study team that future studies can be performed to evaluate for presence of activated T-cells or SARS-COV-2 specific T-cells.”
Submission Date: 24 April 2021
Date of this review: 28 May 2021 15:15:38